# Dalpiciclib combined with pyrotinib and endocrine therapy in women with ER-positive, HER2-positive advanced breast cancer: A prospective, multicenter, single-arm, phase 2 trial

Jian Zhang[1,2☯]*, Yanchun Meng[1,2☯], Biyun Wang[1,2], Xinhong Wu[3], Hongmei Zheng[3], Jing Hu[4], Wei Liu[4], Wenyan Chen[5], Leiping Wang[1,2], Jun Cao[1,2], Zhonghua Tao[1,2], Ting Li[1,2], Sujie Ni[6], Zhengyan Yu[7], Lichun Sun[4], Yun Wang[5], Qiang Peng[5], Song Wang[7], Xin Hu[8], Jianfei Wang[9], Yijia Wu[9], Xichun Hu[1,2]*

1 Department of Medical Oncology, Fudan University Shanghai Cancer Center, Shanghai, China, 2 Department of Oncology, Shanghai Medical College, Fudan University, Shanghai, China, 3 Department of Breast Oncology, Hubei Cancer Hospital, Wuhan, China, 4 Department of Medical Oncology, The Affiliated Tumour Hospital of Harbin Medical University, Harbin, China, 5 Department of Medical Oncology, Nanchang People's Hospital, Nanchang, China, 6 Department of Medical Oncology, Affiliated Hospital of Nantong University, Nantong, China, 7 Department of Medical Oncology, Tumour Hospital of Mudanjiang City, Mudanjiang, China, 8 Precision Cancer Medicine Center, Fudan University Shanghai Cancer Center, Shanghai, China, 9 Jiangsu Hengrui Pharmaceuticals Co., Ltd., Shanghai, China

☯ These authors contributed equally to this work.
* syner2000@163.com (JZ); xchu2009@hotmail.com (XH)

## Abstract

### Background

Combination of HER2-targeted therapy and endocrine therapy offers a more tolerable alternative to HER2-targeted chemotherapy regimens for estrogen receptor (ER)-positive, HER2-positive advanced breast cancer (ABC), but with compromised efficacy. The addition of cyclin-dependent kinase 4/6 (CDK4/6) inhibition may enhance the durability of anti-tumor responses, offering a potential chemotherapy-sparing alternative, although its role in the frontline setting remains uncertain. We performed a multicenter, single-arm, phase 2 clinical trial (PLEASURABLE) to assess the activity and safety of combining dalpiciclib with pyrotinib and endocrine therapy in patients receiving first- or second-line treatment for ER-positive and HER2-positive ABC.

### Methods and findings

We enrolled patients with ER-positive and HER2-positive ABC between August 1, 2019, and November 28, 2022 in this prospective, investigator-initiated trial conducted at six centers in China. Patients received dalpiciclib (125 mg once daily, on days 1–21 of each 28-day cycle) and pyrotinib (320 mg once daily) plus endocrine therapy determined by the physician's choice (letrozole or fulvestrant). The primary endpoint was the objective response rate (ORR), while secondary endpoints included progression-free

**Data availability statement:** All relevant de-identified data supporting the findings of this study are included within the manuscript and its Supporting Information files. The raw clinical data are protected due to privacy laws and unavailable. The trial protocol is also included in the Supplementary Information.

**Funding:** This study was supported by the National Natural Science Foundation of China (https://www.nsfc.gov.cn/) (grant no. 82373359 to JZ), the Project of Shanghai Municipal Health Commission (https://wsjkw.sh.gov.cn/) (grant no. 202140397 to JZ), the CSCO Cancer Research Fund (https://xisike.csco.org.cn/cn/index.aspx) (grant no. Y-HR2020MS-0298 and Y-pierrefabre202102-0066 to JZ), the Chinese Young Breast Experts Research project (grant no. CYBER-2021-001 to JZ), and the Beijing Science and Technology Innovation Medical Development Foundation Key Project (grant no. KC2022-ZZ-0091-6 to JZ). The funders had no role in the study design, data collection and analysis, decision to publish, or preparation of the manuscript.

**Competing interests:** I have read the journal's policy and the authors of this manuscript have the following competing interests: J.F.W. and Y.J.W. were employees of Jiangsu Hengrui Pharmaceuticals Co., Ltd (Shanghai, China) during the study period. All the interests have no financial stake in the results of the study. The remaining authors have no conflicts of interest to declare.

**Abbreviations:** ABC, advanced breast cancer; AI, aromatase inhibitor; CBR, clinical benefit rate; CDK4/6, Cyclin-dependent kinase 4/6; CI, confidence interval; ctDNA, circulating tumor DNA; DCR, disease control rate; DOR, duration of response; ECOG, Eastern Cooperative Oncology Group; ER, estrogen receptor; HR, hormone receptor; ILD, interstitial lung disease; IQR, interquartile range; mITT, modified intent-to-treat; NGS, next-generation sequencing; ORR, objective response rate; OS, overall survival; PFS, progression-free survival; PK, pharmacokinetics; PR, progesterone receptor; TRAE, treatment-related adverse event

survival (PFS), duration of response (DOR), disease control rate (DCR), clinical benefit rate (CBR), safety, plasma pharmacokinetics (PK), and biomarker analysis. Efficacy was analyzed in the modified intention-to-treat population, comprising patients with at least one post-baseline tumor assessment. Safety was assessed in all patients who received at least one dose. A total of 51 patients were screened, and 48 were evaluable (median age was 52.5 years [range, 29–74]); 31 (64.6%) had prior HER2-target therapy, and 37 (77.1%) had received prior endocrine therapy. Thirty (62.5%) and 18 (37.5%) patients received the study treatment as first- and second-line HER2-targeted treatment for ABC, respectively. As of the data cutoff on December 11, 2024, six patients were lost to follow-up, and the median follow-up was 27.3 months (interquartile range, 24.8–30.5). The investigator confirmed ORR was 70.2% (95% CI [55.1, 82.7]), with a DCR of 100% (95% CI [92.5, 100]) and a CBR of 87.2% (95% CI [74.3, 95.2]). The median PFS was 22.0 months (95% CI [16.6, 26.6]), and the median DOR was 22.3 months (95% CI [16.4, 26.9]). No new safety signals were observed, and no treatment-related deaths occurred with only one (2.1%) grade 1 alopecia and no interstitial lung disease. Grade 3 or 4 treatment-related adverse events occurred in 68.8% and 12.5% of patients, respectively, mostly myelosuppression. PK analysis showed no major drug accumulation for dalpiciclib or pyrotinib over the treatment period. Of interest, no objective response was observed in three patients with detected *BRCA* mutations (n=2) or increased $^{68}$Ga-HER2 affibody uptake over the initial two cycles (n=2). The findings of this study should be interpreted with caution due to the limited patient cohort and sample size in exploratory analyses.

## Conclusions

The non-intravenous, chemotherapy-sparing combination of dalpiciclib, pyrotinib, and endocrine therapy demonstrated anti-tumor activity with a manageable safety profile in the frontline treatment of ER-positive, HER2-positive ABC, supporting its further evaluation as a potential alternative.

## Trial registration

ClinicalTrials.gov Identifier: NCT03772353

## Author summary
### Why Was This Study Done?

- The integration of HER2-targeted therapies with chemotherapy is the standard treatment for HER2-positive advanced breast cancer (ABC). However, not all patients are suitable for chemotherapy.

- HER2-targeted therapy combined with endocrine therapy is a treatment option for ER-positive, HER2-positive ABC, offering lower toxicity but with compromised efficacy.

- The addition of cyclin-dependent kinase 4/6 (CDK4/6) inhibition to HER2-targeted therapy and endocrine therapy has shown efficacy in ER-positive/HER2-positive later-line ABC, providing a potential chemotherapy-free alternative.

- To the best of our knowledge, the role of CDK4/6 inhibition in combination with HER2-targeted and endocrine therapy in the frontline setting remains uncertain.

### What Did the Researchers Do and Find?

- We recruited 48 patients with ER-positive, HER2-positive ABC across multiple centers to receive a combination of dalpiciclib, pyrotinib, and endocrine therapy as first- or second-line treatment.

- 70 % (33 of 47) of patients achieved an objective response, with a median progression-free survival of 22.0 months.

- The safety profile was manageable, with no new safety signals observed.

- Exploratory analysis suggested a potential lack of response in patients with increased [68]Ga-HER2 affibody uptake or *BRCA* mutation.

### What Do These Findings Mean?

- The combination of dalpiciclib, pyrotinib, and endocrine therapy appears to be clinically active in ER-positive, HER2-positive advanced breast cancer, with a manageable safety profile, albeit in a relatively small patient cohort.

- The potential effectiveness of CDK4/6 inhibitors dalpiciclib in this study suggests they might potentially be an effective chemotherapy-free option for patients with ER-positive, HER2-positive breast cancer. However, since this study included a small number of patients and did not compare this treatment with standard therapies, more research is needed to confirm the findings.

## Introduction

The combination of HER2-targeted therapies, particularly trastuzumab-pertuzumab, with standard chemotherapy has revolutionized the treatment of HER2-positive advanced breast cancer (ABC), significantly improving patient outcomes, and representing a widely adopted standard of care regardless of estrogen receptor (ER) status [1–3]. Approximately 50% of HER2-positive breast cancers are ER-positive, exhibiting distinct treatment responses and prognoses from ER-negative cases [4–7], partly influenced by bidirectional crosstalk between the HER2 and ER pathways [8]. Additionally, comorbidities, personal preferences, or performance status may preclude the use of chemotherapy in some patients. International guidelines recommend that anti-HER2 therapy combined with an endocrine agent be considered for ER-positive, HER2-positive ABC [9,10]. Nevertheless, these approaches offer compromised efficacy relative to current standard treatments [11,12].

Endocrine-based therapy is preferred over chemotherapy for hormone receptor [(HR, ER and/or progesterone receptor (PR)] -positive, HER2-negative ABC due to its effectiveness and lower toxicity [13]. Cyclin-dependent kinase 4/6 (CDK4/6) inhibitors, effectively halting the cell cycle and slowing cancer cell growth by preventing CDK4/6 from binding to cyclin D and stopping the phosphorylation of the retinoblastoma protein, combined endocrine therapy have made significant survival improvement in HR-positive, HER2-negative ABC [14]. Additionally, CDK4/6 inhibitor reduces TSC2 phosphorylation, which partially attenuates mTORC1 activity and relieves feedback inhibition of upstream EGFR family kinases, resensitizing tumors to EGFR/HER2 blockade [15]. The monarcHER [16] and PATRICIA [17] trial indicates that inhibiting CDK4/6 could bolster the longevity of anti-tumor effects elicited by the concurrent use of HER2-targeted. Long-term follow-up data from the monarcHER trial indicated that triple targeting of CDK4/6, HER2, and ER resulted in numerically higher overall

survival (OS) in ER-positive, HER2-positive ABC patients compared to those treated with standard regimens, following an extensive treatment strategy [18]. The use of triple-target regimen represents an alternative strategy and may provide a chemotherapy-sparing alternative, although its role in the frontline setting remains uncertain.

Dalpiciclib (SHR6390) is a highly selective CDK4/6 inhibitor, which has been proven effective in the DAWNA-1 [19] and DAWNA-2 [20] trials, and approved for the treatment of HR-positive, HER2-negative ABC in China. Pyrotinib, an irreversible pan-ErbB receptor tyrosine kinase inhibitor [21], has also been authorized in China for the management of HER2-positive ABC by PHEOBE and PHENIX trials [21,22], and recently approved for first line in advanced stages, based on efficacy and manageable toxicity in a randomized phase 3 PHILA trial [23]. Notably, preclinical studies have indicated that the combination of dalpiciclib and pyrotinib offers synergistic benefits [24,25]. Early-phase clinical trial (LORDSHIPS) focused on the dose exploration of dalpiciclib, pyrotinib, and letrozole has shown a median progression-free survival (PFS) of 11.3 months and a 66.7% response rate in the frontline setting for patients with ER-positive, HER2-positive ABC [26].

Therefore, we conducted a dose-expansion phase 2 study to evaluate the efficacy and safety of dalpiciclib plus pyrotinib combined with endocrine therapy as a chemotherapy-sparing treatment option for patients with ER-positive, HER2-positive ABC in the frontline setting.

## Methods

### Study design

The PLEASURABLE trial (YBCSG-20-01), initiated by investigators, was a prospective, single-arm, multicenter, phase 2 study conducted at six academic tertiary hospitals in China. Participants were recruited via physician referrals, with eligible patients identified during routine clinical visits or hospital stays. The study was approved from the Medical Ethics Committees of Fudan University Shanghai Cancer Center, Hubei Cancer Hospital, the Affiliated Tumour Hospital of Harbin Medical University, Nanchang People's Hospital, Affiliated Hospital of Nantong University, and Tumour Hospital of Mudanjiang City (S1 Text). The trial was designed and conducted in accordance with the Declaration of Helsinki and the Guidelines for Good Clinical Practice. All participants provided written informed consent before enrollment. This trial was registered on December 11, 2018, at www.clinicaltrials.gov (NCT03772353). Phase 1b of this study has been published [26]. Phase 2 study initially commenced enrollment based on the original protocol approved by the ethics committee (S1 Protocol). Some patients had previously received aromatase inhibitors (AIs) in the adjuvant or first-line setting and experienced disease progression, indicating endocrine resistance. Consequently, continued use of letrozole, as specified in the original protocol, was deemed inappropriate for these patients. Fulvestrant was, therefore, introduced as an alternative endocrine partner in later protocol versions. During the enrollment process, the protocol was amended to Version 3.0 (December 21, 2022) (S1 Protocol), which received ethics committee approval. This version also included an adjustment to the planned sample size. A summary of the protocol amendments is provided in the supplementary material (S2 Text). All participants enrolled prior to the implementation of Version 3.0 met the inclusion and exclusion criteria of that version, except for one participant who did not meet the enrollment criteria. This participant did not meet the original hemoglobin criteria but, after enrollment and symptomatic treatment, reached the required level and was included in the safety and efficacy analysis sets. The study was reported in accordance with the CONSORT guidelines. The completed CONSORT checklist is provided as supporting information (S1 CONSORT Checklist).

### Participants

Female patients aged 18–75 years, regardless of menopausal status (premenopausal or perimenopausal patients received luteinizing hormone–releasing hormone agonists), with a confirmed diagnosis of ER-positive, HER2-positive breast cancer, as well as unresectable, locally advanced, recurrent, or metastatic disease were eligible for participation in

this trial. Untreated but asymptomatic brain metastases were allowed for inclusion. The patient had received at most one systemic treatment with trastuzumab, including anti-HER2 antibody–drug conjugate, for recurrent or metastatic breast cancer. Prior anti-HER2 tyrosine kinase inhibitors therapy was either absent or not shown to be failed. Previous endocrine therapy was allowed. Other eligibility criteria included having at least one measurable extracranial lesion following RECIST 1.1 guidelines, an Eastern Cooperative Oncology Group (ECOG) performance status of 0–1, and adequate organ function. Key exclusion criteria included known meningeal metastasis or active brain parenchymal metastasis that were symptomatic or required steroids. Previous CDK4/6 inhibitor treatment and concurrent conditions affecting safety or study completion assessed by investigators were also exclusion criteria.

## Interventions and procedures

The recommended doses of dalpiciclib and pyrotinib were determined from the dose-finding phase 1b trial (LORDSHIPS [26]). Patients received dalpiciclib (125 mg, once daily, on days 1–21 of each 28-day cycle) and pyrotinib (320 mg, once daily) plus endocrine therapy determined by the physician's choice. For patients who have shown resistance to AIs, defined as recurrence during or within 1 year after adjuvant AI treatment or those who experienced disease progression while receiving AIs in the recurrence and metastasis stage, fulvestrant (500 mg intramuscularly on days 1 and 15 of cycle 1 and then on day 1 once every 4 weeks) would be administered upon enrollment. Otherwise, letrozole (2.5 mg, once daily) would be initiated. Treatment was maintained until disease progression according to RECIST 1.1, unacceptable toxicity, withdrawal of consent by the patient or physician at any time. There were no pre-medications required for this regimen. All treatments were administered as outlined in the study protocol. The trial protocol can be found in the Supplement (S1 Protocol).

## Outcomes

The primary endpoint was the ORR, defined as the proportion of patients with the best response of confirmed complete or partial response according to RECIST 1.1, as evaluated by the investigator. Secondary endpoints included DOR, DCR, clinical benefit rate (CBR, proportion of complete response, partial response, or stable disease for at least 24 weeks), PFS, safety, plasma pharmacokinetics (PK) and biomarker analysis.

Tumor assessments were done by contrast-enhanced computed tomography or magnetic resonance imaging at baseline and every two cycles for the first six cycles, and every three cycles thereafter. Complete or partial response would be confirmed 4–6 weeks by the investigators per RECIST v1.1. Adverse events were monitored at each patient follow-up and graded according to the National Cancer Institute Common Terminology Criteria version 4.03.

In terms of PK, plasma samples were collected during the first cycle on day 21 at the following time points relative to pyrotinib administration: samples were obtained at predose and at 2, 4, 6, 12, and 24 hours after the dose. Additionally, plasma samples were collected on cycle 3, 5, 8, and 12 day 22, all within 0.5 h before pyrotinib administration. Plasma concentrations of dalpiciclib, pyrotinib, and letrozole were measured by using a validated liquid chromatography-tandem mass spectrometry method.

Biomarker analysis included $^{68}$Ga-HER2 affibody with PET-CT and [18F]2-fluoro-2-deoxy-D-glucose (FDG)–PET/CT examinations, both performed at baseline and after two treatment cycles. HER2-PET and FDG-PET imaging were conducted under fasting conditions. Patients received an intravenous injection of $^{68}$Ga-HER2 affibody or F-18-FDG, followed by a 60-minute resting period post-injection. Whole-body PET-CT scans to assess tracer uptake in primary and metastatic lesions. Tissue-derived DNA and circulating tumor DNA (ctDNA) were analyzed by next-generation sequencing (NGS) using 539- or 141-gene panels to detect somatic and germline mutations based on baseline samples. Although additional samples were collected after two treatment cycles or at disease progression in some patients, these data are not included in the current analysis due to resource constraints.

## Statistical analysis

Expecting a 60% efficacy in the phase 2 study, derived from the phase 1b study (LORDSHIPS) with an ORR of 66.7%, a sample size of 41 patients was required to achieve a half-width of 15% for a two-sided 95% confidence interval (CI). Considering a dropout rate of 15%, a total of 48 patients were required. Four subjects from phase 1b with the recommended phase 2 dose (RP2D) can be included in the phase 2 study analysis, leaving 44 subjects to be enrolled. Analysis of efficacy in the modified intent-to-treat (mITT) population which included eligible participants who underwent at least one post-baseline tumor assessment, while safety analysis encompassed all patients who received at least one dose of the study treatment. The primary endpoint ORR was estimated with a 95% CI using the Clopper–Pearson method. Additionally, DCR and CBR were estimated, providing their representative 95% CIs. PFS and the survival rate, along with their respective 95% CIs were estimated through the Kaplan–Meier method. SAS (version 9.2) or above (North Carolina, USA) was utilized for all statistical analyses except pharmacokinetic analysis (Phoenix WinNonlin 8.1).

## Results

### Patient disposition and baseline characteristics

Between August 1, 2019, and November 28, 2022, a total of 51 patients were screened, and 48 were enrolled (median age was 52.5 years [range, 29–74]) and received study treatment (Fig 1). Efficacy was assessed in the mITT population (n = 47), excluding one patient—who withdrew after one cycle for liver surgery without post-treatment imaging—while still included in the safety analysis. Baseline demographics and disease characteristics are given in Table 1. Among all patients, 30 (62.5%) and 18 (37.5%) received the study treatment as first- and second-line HER2-targeted treatment for ABC, respectively. Thirty-one (64.6%) had received previous HER2-target therapy and 37 (77.1%) patients had received previous endocrine therapy in any setting. Visceral metastases were present in 77.1% of patients, with 16.7% with asymptomatic untreated brain metastases.

### Efficacy

As of the data cutoff on December 11, 2024, six patients were lost to follow-up and the median follow-up was 27.3 months (interquartile range [IQR] 24.8–30.5). The investigator confirmed objective response rate (ORR, RECIST version 1.1), the primary endpoint, was 70.2% (95% CI [55.1, 82.7]; 33 of 47 patients) (Fig 2). The disease control rate (DCR) was 100% (95% CI [92.5, 100]). One patient (2.1%) achieved a confirmed complete response, 32 patients (68.1%) achieved a confirmed partial response and eight patients (17.0%) achieved a stable disease for ≥ 24 weeks (Fig 2), resulting in a CBR of 87.2% (95% CI [74.3, 95.2]). The median PFS was 22.0 months (95% CI [16.6, 26.6]) (Fig 3). The median duration of response (DOR) was 22.3 months (95% CI [16.4, 26.9]) with ongoing response in 33.3% of patients (Fig 2).

OS was evaluated in a post-hoc analysis. As of the data cutoff, OS data were immature, with a total of seven events observed. The estimated 48-month OS rates were 82.6% (95% CI [66.3, 91.5]) (S1 Fig).

In terms of subgroups, post-hoc analyses revealed that the confirmed ORR in patients with first- and second-line HER2-targeted therapy was 79.3% (95% CI [60.3, 90.2]) and 55.6% (95% CI [30.8, 78.5]), respectively (Figs 2 and S2). The overall response was 87.5% (95% CI [61.7, 98.5]) in trastuzumab-naïve patients and 61.3% (95% CI [42.2, 78.2]) in trastuzumab-treated patients, and 81.5% (95% CI [61.9, 93.7]) in patients receiving endocrine therapy with letrozole and 55.0% (95% CI [31.5, 76.9]) in patients receiving fulvestrant (S2 Fig). The median PFS in patients with first- and second-line HER2-targeted therapy was 22.3 months (95% CI [16.6, 26.6]) and 20.1 months (95% CI [7.6, 44.1]), respectively (Fig 3). The PFS was 20.9 (95% CI [11.2, not reached]) months in trastuzumab-naïve patients and was 22.0 months (95% CI [11.3, 30.5]) in trastuzumab-treated patients (S3 Fig), and 22.3 months (95% CI [19.2, 44.1]) in patients receiving endocrine therapy with letrozole and 15.8 months (95% CI [5.7, 25.8]) in patients receiving fulvestrant (S4 Fig).

PLOS Medicine

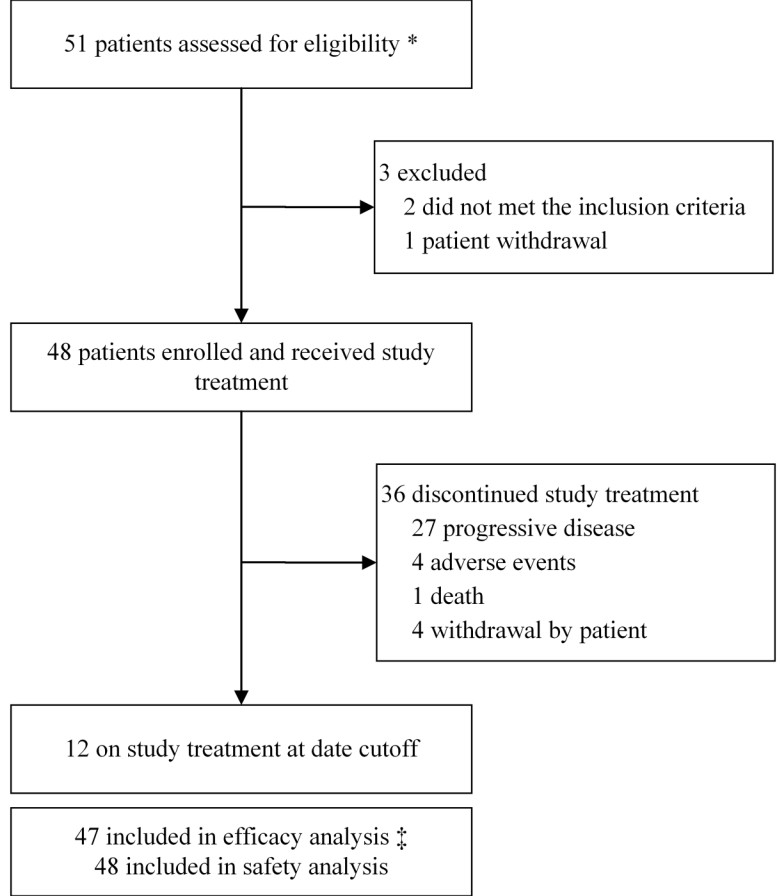

**Fig 1. Trial profile.** * Inclueds four subjects from phase 1b with the recommended phase 2 dose. ‡ The mITT population comprised 47 patients, one patient was withdrawn from the study for liver surgery without imaging evaluation after one treatment cycle, and excluded from the efficacy analysis set. Two patients with only intracranial progression during the study period resumed study treatment, with one undergoing local stereotactic radiosurgery at the discretion of the patients and treating physicians.

A post-hoc analysis revealed that eight (16.7%) out of 48 patients with asymptomatic untreated brain metastases at study entry achieved an ORR of 62.5% (95% CI [24.5, 91.5]) and a median PFS of 17.4 months (95% CI [5.5, 25.8]). These numbers are too small to draw any definitive conclusions (S5 Fig).

**Safety**

The safety population consisted of 48 patients (Table 2). All the 48 patients who received at least one dose of the study drug had experienced at least one treatment-related adverse event (TRAE) during the study treatment. The most common TRAEs included neutropenia (97.9%), leukopenia (95.8%), diarrhea (91.7%), anemia (87.5%), oral mucositis (70.8%), and thrombocytopenia (50.0%). Grade 3 and 4 TRAEs occurred in 33 (68.8%) and 6 (12.5%) patients, respectively, primarily myelosuppression. The most common grade 3 TRAEs were neutropenia (60.4%), leukopenia (56.3%), and anemia (10.4%). Neutropenia and thrombocytopenia were reported grade 4 TRAE, occurring in 12.2% and 2.1% of patients, respectively. Notably, no interstitial lung disease (ILD) was observed. Alopecia was rare, reported as grade 1 in one patient (2.1%). Four patients (8.3%) experienced treatment-related serious adverse events, requiring hospitalization, including one grade 3 leukopenia, one grade 3 thrombocytopenia, one

**Table 1. Baseline characteristics of patients.**

| | Total [a] (n = 48) | First line (n = 30) | Second line (n = 18) |
|---|---|---|---|
| Age | | | |
| Median (range), years | 52.5 (29-74) | 53.5 (29-73) | 48.5 (31-74) |
| ECOG performance states, n (%) | | | |
| 0 | 13 (27.1) | 7 (23.3) | 6 (33.3) |
| 1 | 35 (72.9) | 23 (76.7) | 12 (66.7) |
| Menopausal status, n (%) | | | |
| Pre- or perimenopausal | 13 (27.1) | 8 (26.7) | 5 (27.8) |
| Postmenopausal | 35 (72.9) | 22 (73.3) | 13 (72.2) |
| Disease status, n (%) | | | |
| Metastatic, de novo | 9 (18.8) | 9 (30.0) | 0 |
| Metastatic, recurrent | 39 (81.2) | 21 (70.0) | 18 (100) |
| Endocrine therapy determined by physician's choice [b], n (%) | | | |
| Letrozole [c] | 28 (58.3) | 18 (55.6) | 10 (55.6) |
| Fulvestrant | 20 (41.7) | 12 (44.4) | 8 (44.4) |
| Hormone receptor status, n (%) | | | |
| ER+ and PR+ [d] | 33 (68.8) | 21 (70.0) | 12 (66.7) |
| ER+ and PR- | 15 (31.2) | 9 (30.0) | 6 (33.3) |
| Estrogen receptor status, n (%) | | | |
| ER ≥ 50% | 39 (81.2) | 24 (80.0) | 15 (83.3) |
| 10% ≤ ER < 50% | 8 (16.7) | 6 (20.0) | 2 (11.1) |
| 1% ≤ ER < 10% | 1 (2.1) | 0 | 1 (5.6) |
| HER2 status, n (%) | | | |
| IHC 3+ | 28 (58.3) | 18 (60.0) | 10 (55.6) |
| IHC 2+ and FISH + | 20 (41.7) | 12 (40.0) | 8 (44.4) |
| Number of metastatic sites, n (%) | | | |
| ≥ 3 | 22 (45.8) | 12 (40.0) | 10 (55.6) |
| < 3 | 26 (54.2) | 18 (60.0) | 8 (44.4) |
| Metastatic sites at screening, n (%) | | | |
| Visceral | 37 (77.1) | 24 (80.0) | 13 (72.2) |
| Brain | 8 (16.7) | 5 (16.7) | 3 (16.7) |
| Liver | 14 (29.2) | 10 (33.3) | 4 (22.2) |
| Lung | 27 (56.3) | 17 (56.7) | 10 (55.6) |
| Non-visceral | 11 (22.9) | 6 (20.0) | 5 (27.8) |
| Brain metastases at screening, n (%) | | | |
| Received local therapy | 0 | 0 | 0 |
| Did not receive local | 8 (16.7) | 5 (16.7) | 3 (16.7) |
| Previous HER2-target therapy, n (%) | | | |
| None | 17 (35.4) | 17 (56.7) | 0 |
| Trastuzumab | 31 (64.6) | 13 (43.3) | 18 (100) |
| Advanced setting | 15 (31.3) | 0 | 15 (83.3) |
| Neoadjuvant/Adjuvant | 18 (37.5) | 13 (43.3) | 5 (27.8) |
| Pertuzumab | 10 (20.8) | 3 (10) | 7 (38.9) |
| T-DM1 | 1 (2.1) | 1 (3.3) | 0 |

*(Continued)*

**Table 1.** (Continued)

| | Total [a] (n=48) | First line (n=30) | Second line (n=18) |
|---|---|---|---|
| Previous endocrine therapy, n (%) | | | |
| None | 11 (22.9) | 9 (30.0) | 2 (11.1) |
| Yes | 37 (77.1) | 21 (70.0) | 16 (88.9) |
| Aromatase inhibitors | 27 (56.3) | 15 (50.0) | 12 (66.7) |
| Advanced setting | 10 (20.8) | 4 (13.3) | 6 (33.3) |
| Neoadjuvant/Adjuvant | 17 (35.4) | 11 (36.7) | 6 (33.3) |
| SERM | 18 (37.5) | 11 (36.7) | 7 (38.9) |
| Previous chemotherapy, n (%) | | | |
| None | 11 (22.9) | 11 (36.7) | 0 |
| Yes | 37 (77.1) | 19 (63.3) | 18 (100) |
| Anthracycline | 28 (58.3) | 17 (56.7) | 11 (61.1) |
| Taxane | 34 (70.8) | 16 (53.3) | 18 (100) |
| No. of prior systemic treatments | | | |
| Median (range) | 1 (0–3) | 1 (0–3) | 2 (1–3) |
| Prior systemic treatment, n (%) | | | |
| None | 9 (18.8) | 9 (30.0) | 0 |
| Neoadjuvant/adjuvant | 31 (64.6) | 20 (66.7) | 11 (61.1) |
| Advanced setting | 17 (35.4) | 2 (6.7) | 15 (83.3) |

a. Baseline characteristics grouped by line of HER2-targeted therapy. First-line HER2-targeted therapy was defined as having no history of trastuzumab treatment or relapse more than one year after the end of trastuzumab-based adjuvant therapy. Second-line HER2-targeted therapy was defined as relapse during or within one year after the end of adjuvant trastuzumab treatment or progression on first-line trastuzumab treatment for advanced disease.

b. For patients with prior endocrine therapy showing resistance to aromatase inhibitors, defined as recurrence during or within one year after adjuvant aromatase inhibitor treatment, or those who received aromatase inhibitors in the recurrence and metastasis stage with disease progression, fulvestrant would be administered upon enrollment. Otherwise, letrozole treatment would be initiated.

c. Two patients enrolled to continue treatment with letrozole after progressing on aromatase inhibitor therapy in the advanced stage at the discretion of physicians.

d. ER+ defined as ER expression in ≥1% tumor cells; PR+ defined as ER expression in ≥1% tumor cells.

ECOG eastern cooperative oncology group, ER estrogen receptor, FISH fluorescence in situ hybridization, HER2 human epidermal growth factor receptor 2, IHC immunohistochemistry, PR progesterone receptor, SERM selective estrogen receptor modulator, TRAST trastuzumab.

grade 3 hypokalemia, and one grade 3 renal impairment. One death occurred during the study, and two others occurred within 30 days after the last dose following withdrawal from the study; all were considered potentially linked to disease progression that led to a deterioration in general physical health, including one due to brain metastases. No treatment-related deaths occurred.

Dose adjustments were required for 16 (33.3%) patients. Of these, seven required dose reductions for pyrotinib, with six of them adjusted to 240 mg/d, citing reasons like diarrhea, neutropenia and leukopenia, rash, oral mucositis, and renal impairment. In addition, a patient developed unrelated cryptococcal pneumonia and received fluconazole therapy. This prompted the adjustment of the pyrotinib dosage to 80 mg/d due to drug interactions. Fourteen (29.2%) patients required adjustment of the dalpiciclib dose to 100 mg/d, mostly due to neutropenia and leukopenia, with one needing a second adjustment to 75 mg/d. Treatment discontinuation owing to TRAEs occurred in three patients, each attributed to specific adverse events: recurrent rash, oral mucositis, and thrombocytopenia.

**A**

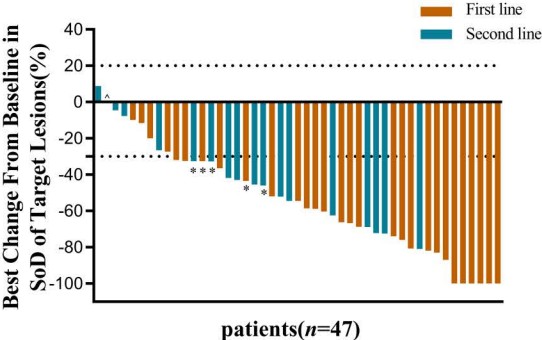

**B**

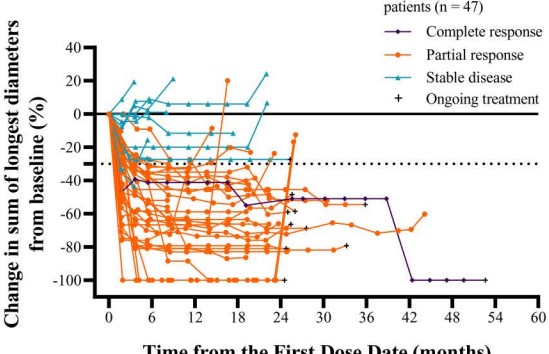

**C**

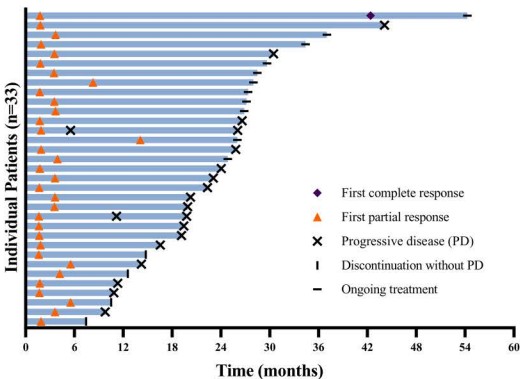

**Fig 2. Tumor response.** (A) Waterfall plot of change from baseline in the sum of the diameters of target lesions by investigator per RECIST v1.1. (B) Change from baseline of the sum of diameters of target lesions and (C) Swimmer plot of duration of response in patients achieving confirmed objective response per RECIST v1.1. * Unconfirmed partial response, three patients with initial partial response exited without subsequent imaging confirmation; the follow-up imaging for the other two cases revealed progressive disease. ^ One patient with second line treatment had no change in targetlesions at the end of the treatment from baseline. × Two patients with only intracranial progression during the study period resumed study treatment, with one undergoing local stereotactic radiosurgery at the discretion of the patients and treating physicians. SoD sum of the diameters, PD progressive disease.

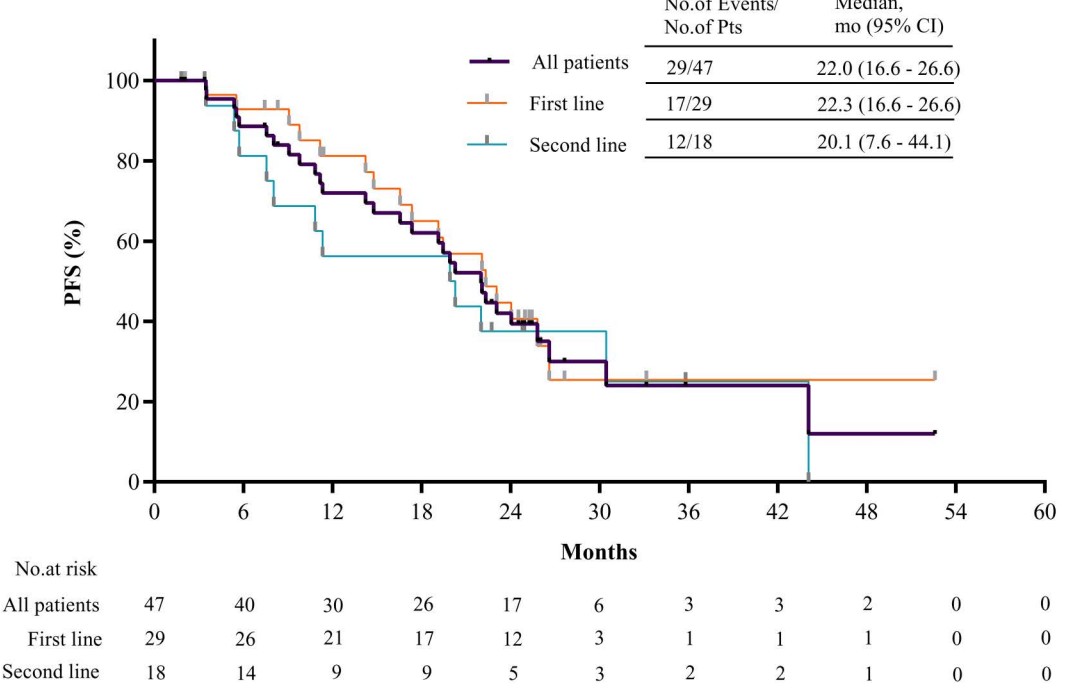

**Fig 3. Progression-free survival.** Kaplan-Meier estimates of progression-free survival in overall efficacy-evaluable population and subgroups of different line of HER2-target therapy. CI confidence interval, PFS progression-free survival.

## Pharmacokinetics analysis

A total of 12 participants provided samples for the PK analysis. The steady-state $C_{max}$ on day 21 of dalpiciclib, pyrotinib, and letrozole was 133, 93.6, and 125 ng/mL, respectively. Similar exposure of dalpiciclib was observed between the initial and subsequent cycles. The exposure of pyrotinib was 1.22- to 1.68-fold greater than initial cycle exposure, as assessed by the $C_{min,ss}$ ratio (R: cycles $C_{through}$ on study day 22 to $C_{through}$ on study cycle 1 day 22) (S1 Table). These results suggest that there is no major accumulation of dalpiciclib and pyrotinib after long-term administration.

## Biomarker analyses

Twenty patients underwent baseline [68]Ga-HER2 and 18F-FDG PET/CT scans. HER2 expression was detected in 16 cases by [68]Ga-HER2 PET/CT scans. The heterogeneity of [68]Ga-HER2 affibody uptake was observed among patients, both at baseline (N = 16, median SUV $_{max}$ 7.5, range, 2.3–34.9) and after 2 cycles of therapy (N = 12, median SUV $_{max}$ 4.0, range, 0–14.9). Among the seven patients with reduced [68]Ga-HER2 affibody uptake, partial response was achieved. Conversely, two patients with elevated [68]Ga-HER2 affibody uptake alongside decreased FDG metabolism exhibited stable disease (S2 Table).

Sixteen patients had baseline ctDNA or tumor samples submitted for central testing. NGS was conducted to detect somatic and germline mutations, involving 10 tissue samples and 10 ctDNA samples (S3 Table). The most frequently mutated genes in patients included *ERBB2* (68.8%), *TP53* (50%), *PIK3CA* (37.5%), *FGFR1* (25%), *MYC* (18.8%), and *BRCA* (12.5%). Regarding the mutation of *ERBB2*, 62.5% showed HER2 amplification, while 6.3% exhibited a missense mutation. Two patients with *BRCA* mutations (one somatic *BRCA1/2* and one germline *BRCA2*) showed no tumor response. Due to the limited sample size, no definitive conclusions can be drawn.

**Table 2. Treatment-related adverse events (n = 48).**

| | Any Grade | Grade 3 | Grade 4 |
|---|---|---|---|
| Any event | 48 (100) | 33 (68.8) | 6 (12.5) |
| Neutropenia | 47 (97.9) | 29 (60.4) | 5 (10.4) |
| Leukopenia | 46 (95.8) | 27 (56.3) | 0 |
| Diarrhea | 44 (91.7) | 1 (2.1) | 0 |
| Anemia | 43 (89.6) | 5 (10.4) | 0 |
| Oral mucositis | 34 (70.8) | 2 (4.2) | 0 |
| Thrombocytopenia | 24 (50.0) | 3 (6.3) | 1 (2.1) |
| Hypokalemia | 23 (47.9) | 4 (8.3) | 0 |
| Creatinine increased | 23 (47.9) | 0 | 0 |
| Rash | 19 (39.6) | 0 | 0 |
| Alanine aminotransferase increased | 18 (37.5) | 1 (2.1) | 0 |
| Aspartate aminotransferase increased | 18 (37.5) | 3 (6.3) | 0 |
| γ-glutamyltransferase increased | 16 (33.3) | 3 (6.3) | 0 |
| Hypophosphatemia | 15 (31.3) | 2 (4.2) | 0 |
| Hyperuricemia | 14 (29.2) | 0 | 0 |
| Urine leukocyte positive | 13 (27.1) | 0 | 0 |
| Electrocardiogram T wave abnormal | 12 (25.0) | 0 | 0 |
| Hypertriglyceridemia | 12 (25.0) | 0 | 0 |
| Hypoalbuminaemia | 11 (22.9) | 0 | 0 |
| Haemorrhoidal haemorrhage | 10 (20.8) | 0 | 0 |
| Hypocalcaemia | 10 (20.8) | 0 | 0 |
| Hyperglycemia | 10 (20.8) | 0 | 0 |
| Haematuria | 9 (18.8) | 0 | 0 |
| Blood alkaline phosphatase increased | 8 (16.7) | 0 | 0 |
| Occult blood positive | 8 (16.7) | 0 | 0 |
| Protein urine present | 8 (16.7) | 0 | 0 |
| Hypomagnesaemia | 7 (14.6) | 0 | 0 |
| Vomiting | 7 (14.6) | 0 | 0 |
| Weight decreased | 6 (12.5) | 0 | 0 |
| Lymphocyte count decreased | 5 (10.4) | 2 (4.2) | 0 |
| Blood bilirubin increased | 5 (10.4) | 0 | 0 |
| Ventricular extrasystoles | 5 (10.4) | 0 | 0 |
| Nausea | 5 (10.4) | 0 | 0 |

Adverse events of all grades occurring in ≥10% of the safety population. Data are n (%), listed by decreasing frequency. Two grade 3 treatment-related adverse events not included in Table 2 were: renal impairment (1 subject, 2.1%) and skin infection (1 subject, 2.1%).

## Discussion

To our knowledge, PLEASURABLE is the first study to explore a chemotherapy-sparing regimen incorporating dalpiciclib, pyrotinib, and endocrine therapy in the frontline treatment of ER-positive, HER2-positive ABC. The observed ORR of 70.2% and median PFS of 22.0 months, along with an acceptable safety profile, suggest the potential clinical activity in patients who are ineligible for or prefer to avoid chemotherapy. However, given the small sample size and the absence of a comparator arm, these findings should be interpreted with caution. Larger randomized trials are needed to confirm the efficacy and generalizability of this approach.

For ER-positive, HER2-positive ABC, the current first-line standard consists of trastuzumab, pertuzumab, and docetaxel, as demonstrated in the CLEOPATRA study (ORR 80.2%; median PFS 18.7 months) [27,28] and confirmed by the PUFFIN trial conducted in China (ORR 79%; median PFS 16.5 months) [3,29]. More recently, pyrotinib combined with trastuzumab and docetaxel was approved in China as the first-line regimen (ORR 83%; median PFS 24.3 months) [23]. In our study, among patients receiving first-line HER2-targeted therapy, 43.3% had prior trastuzumab exposure, and 70% had received endocrine therapy. This subgroup achieved a comparable ORR of 79.3% (95% CI [60.3 to 90.2]) and median PFS of 22.3 months (95% CI [16.6, 26.6]). However, caution is warranted as standard first-line trials also included ER-negative patients, whereas our study had a limited sample size and lacked a control group. For second-line treatment, trastuzumab deruxtecan (T-DXd) is now considered the preferred option (ORR 79%; median PFS 28.8 months) [30], although its use may be restricted by cost and availability. Other approved regimens include pyrotinib plus capecitabine (ORR 67%; median PFS 12.5 months) [21] and trastuzumab emtansine (ORR 43.6%; median PFS 9.6 months) [31]. All patients in our second-line subgroup had progressed on prior trastuzumab, and 88.9% had also received endocrine therapy. The observed ORR was 55.6% (95% CI [30.8, 78.5]), with a median PFS of 20.1 months (95% CI [7.6, 44.1]). Given the small sample size and wide CIs, these findings are preliminary and should be interpreted with caution.

Endocrine therapy combined with HER2-targeted agents offers a chemotherapy-sparing alternative but has shown compromised PFS outcomes (4.8–14.1 months) [11,12,32]. The SYSUCC-002 trial, conducted in China, reported efficacy with first-line trastuzumab plus letrozole (ORR 37.2%; median PFS 19.2 months) [33]. Variable efficacy has also been observed across similar yet distinct regimens. Yan Min and colleagues reported frontline pyrotinib plus dalpiciclib in the HR-positive subgroup (ORR 55.6%; median PFS 9.1 months) [34], while Ouyang Quchang and colleagues documented outcomes with pyrotinib plus letrozole in first-line setting (ORR 64.2%; median PFS 13.7 months) [35]. Disrupting HER2-ER crosstalk may be effective only in a subset of patients, and activation of alternative tumor survival pathways—such as the cyclin D1–CDK4/6 axis—may limit efficacy [15]. In our study, concurrent inhibition of HER2, ER, and CDK4/6 may have contributed to the observed clinical activity. These findings are preliminary and should be interpreted with caution. In patients with HER2-positive brain metastases, Wang Shusen and colleagues reported a median PFS of 10.6 months with a combination of trastuzumab, pyrotinib, palbociclib, and endocrine therapy [36]. Considering the differences in baseline characteristics and prior treatments, as well as the limited number of patients with brain metastases in our study, definitive conclusions cannot be drawn. Among the endocrine backbones, the letrozole-based regimen yielded numerically higher ORR and PFS than the fulvestrant-based approach. This difference may reflect the fulvestrant subgroup's composition, which likely included patients with prior AI resistance and a higher proportion undergoing second-line anti-HER2 treatments. Nonetheless, these are subgroup findings and not powered for formal comparison; further investigation is warranted.

The combination of dalpiciclib and pyrotinib with endocrine therapy demonstrated a manageable safety profile, with no new or unexpected safety signals. The most common grade 3–4 adverse events related to dalpiciclib were hematologic toxicities, primarily neutropenia and leukopenia, which were effectively managed through dose modifications and granulocyte colony-stimulating factor support. No treatment discontinuation occurred due to neutropenia. The incidence of grade 3–4 neutropenia and leukopenia was numerically lower than in those reported in DAWNA-1 [19] and DAWNA-2 [20]. Diarrhea, the most common pyrotinib-related toxicity, also occurred with a numerically lower incidence of grade ≥3 events compared to pyrotinib plus chemotherapy in the PHOEBE [21] and PHALA trials [23], potentially due to the lower dosing regimen selected based on phase 1b dose exploration. Although oral mucositis occurred at a numerically higher incidence than previously reported for pyrotinib and was rarely observed with dalpiciclib [19–21], it remained manageable with symptomatic treatment. Notably, ILD was not observed in this study, contrasting with the known pulmonary risks of HER2-targeted antibody-drug conjugates such as trastuzumab deruxtecan [30]. Alopecia, a common toxicity associated with chemotherapy-based regimens, was also rare, with only one case of grade 1 alopecia (2.1%). Other gastrointestinal adverse event (e.g., nausea, vomiting, and constipation), and nervous system toxicities (e.g., headache and peripheral

neuropathy) were also numerically less frequent than those reported with HER2-targeted therapy combined with chemotherapy [2]. However, the limited sample size and absence of a direct comparison with chemotherapy-based regimens preclude definitive conclusions.

The pharmacokinetic profile of dalpiciclib in combination was similar to that observed with monotherapy [37]. Pyrotinib exposure was numerically lower than with monotherapy [38], possibly due to interindividual variability and diarrhea. Overall, these results suggest that dalpiciclib does not impact the PK of pyrotinib or letrozole, and its own exposure is not impacted by combination use. With repeated dosing, dalpiciclib exposure remained stable, while pyrotinib exposure was numerically lower than with monotherapy, with a slight increase observed over treatment cycles. No drug accumulation was observed.

Radionuclide imaging allows real-time, non-invasive assessment of HER2 expression across tumors [39]. In our study, a reduction in $^{68}$Ga-HER2 affibody uptake after two cycles correlated with decreased FDG metabolism. Conversely, elevated uptake of $^{68}$Ga-HER2 affibody in two cases may indicate a lower likelihood of benefit from the current targeted therapy, although a definitive conclusion cannot be drawn due to the limited sample size. While evidence remains limited, co-occurring *BRCA* mutations and *HER2* positivity may be associated with poorer outcomes [40]; in our study, both somatic and germline *BRCA* variants detected at baseline were linked to non-response.

Limitations exist in this study. It was a single-arm trial with a small sample size, limiting statistical power and generalizability. The absence of a control group prevents assessment of individual treatment contributions. Most patient (62.5%) received the regimen as first-line treatment without trastuzumab, deviating from the standard approach; whether the addition of trastuzumab is essential in this setting requires further study. Quality-of-life data were not collected, limiting evaluation of patient-reported outcomes. Biomarker analyses were exploratory and limited by small sample size: $^{68}$Ga-HER2 PET/CT was performed in 20 patients and ctDNA in 16, with no post-treatment or progression samples due to funding constraints. These limitations restrict interpretation of molecular responses and resistance mechanisms. The findings are hypothesis-generating and require validation in larger studies.

The non-intravenous, chemotherapy-sparing combination of dalpiciclib, pyrotinib, and endocrine therapy demonstrated anti-tumor activity and manageable toxicity in the frontline treatment of ER-positive, HER2-positive ABC, supporting its further evaluation as a potential alternative. These findings outline a prospective path for further research on CDK4/6 inhibitors in the ER-positive, HER2-positive breast cancer.

## Supporting information

**S1 Fig. Overall survival.**
(DOCX)

**S2 Fig. Exploratory subgroup analyses of ORR by baseline factors.**
(DOCX)

**S3 Fig. Progression-free survival in subgroups of trastuzumab-naïve/-treated therapy.**
(DOCX)

**S4 Fig. Progression-free survival in subgroups of different endocrine therapy.**
(DOCX)

**S5 Fig. Progression-free survival in patients with asymptomatic untreated brain metastases at baseline.**
(DOCX)

**S1 Table. Pharmacokinetic.**
(DOCX)

**S2 Table. Results of $^{68}$Ga-HER2 and 18F-FDG PET/CT.**
(DOCX)

**S3 Table. Results of NGS in baseline.**
(DOCX)

**S1 Protocol. Trial protocol.**
(PDF)

**S1 Text. Ethics approval.**
(DOCX)

**S2 Text. Summary of protocol amendments.**
(DOCX)

**S1 CONSORT Checklist. CONSORT 2010 checklist of information to include when reporting a randomized trial.**
(DOCX)

## Acknowledgments

We would like to acknowledge the patients and their families, the study investigators, and the clinical site staff. The study drugs—dalpiciclib, pyrotinib, and letrozole—were provided by Jiangsu Hengrui Pharmaceuticals Co., and fulvestrant was provided by Chia Tai Tianqing Pharmaceutical Group Co.

## Author contributions

**Conceptualization:** Jian Zhang, Xichun Hu.

**Data curation:** Yanchun Meng, Biyun Wang, Jun Cao, Ting Li, Sujie Ni, Lichun Sun, Yun Wang, Qiang Peng, Song Wang, Xin Hu, Jianfei Wang, Yijia Wu.

**Formal analysis:** Jian Zhang, Yanchun Meng.

**Funding acquisition:** Jian Zhang.

**Investigation:** Jian Zhang, Yanchun Meng, Biyun Wang, Xinhong Wu, Hongmei Zheng, Jing Hu, Wei Liu, Wenyan Chen, Leiping Wang, Zhonghua Tao, Sujie Ni, Zhengyan Yu, Xichun Hu.

**Methodology:** Jian Zhang.

**Project administration:** Jian Zhang, Yanchun Meng.

**Resources:** Jian Zhang.

**Supervision:** Jian Zhang, Xichun Hu.

**Writing—original draft:** Yanchun Meng.

**Writing—review & editing:** Jian Zhang.

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
