## [Editor Report · Decision Letter 0]

4 Nov 2024

Dear Dr Zhang, 

Thank you for submitting your manuscript entitled "Dalpiciclib combined with pyrotinib and endocrine therapy in women with ER-positive, HER2-positive advanced breast cancer (PLEASURABLE): a multicentre, single-arm, phase 2 trial" for consideration by PLOS Medicine.

Your manuscript has now been evaluated by the PLOS Medicine editorial staff and I am writing to let you know that we would like to send your submission out for external peer review.

We would like to thank you for providing the original documents and the document containing the updated methods section by email. Please make sure to upload the relevant documents (i.e. original and final protocol in English and Chinese) when resubmitting your manuscript.

Please re-submit your manuscript within two working days, i.e. by Nov 06 2024.

Feel free to email me at atosun@plos.org or us at plosmedicine@plos.org if you have any queries relating to your submission.

Kind regards,

Alexandra Tosun, PhD

Associate Editor

PLOS Medicine

---

## [Decision Letter · Decision Letter 1]

11 Feb 2025

Dear Dr Zhang,

Many thanks for submitting your manuscript "Dalpiciclib combined with pyrotinib and endocrine therapy in women with ER-positive, HER2-positive advanced breast cancer (PLEASURABLE): a multicentre, single-arm, phase 2 trial" (PMEDICINE-D-24-03621R1) to PLOS Medicine. The paper has been reviewed by subject experts and a statistician; their comments are included below and can also be accessed here: [LINK]

At present, we feel that it is not clear what the research question and aim of your study is/was. Given that PLOS Medicine publishes articles for a wide audience, it is necessary to clearly state the need for your study in the Abstract Background section. In the Introduction, you should address previous research and explain the need for and potential importance of your study (How is it different from previous trials?). You should also indicate how your study is novel, how you have determined this, and what may be novel about the treatment regimen chosen. In addition, with a global audience in mind, we feel that it is currently not clear what the standard of care is in China and how this regimen compares to it (e.g. what are the potential advantages and disadvantages?). You should also provide details on whether the standard of care varies between institutions in China and how the standard of care differs from other settings (e.g. Europe, USA).

As you will see, the reviewers also provided mixed feedback on the manuscript. After discussing the paper with the editorial team, I'm pleased to invite you to revise the paper in response to the reviewers' and editorial comments. We plan to send the revised paper to some or all of the original reviewers, and we cannot provide any guarantees at this stage regarding publication.

We ask that you submit your revision by Mar 04 2025. However, if this deadline is not feasible, please contact me by email, and we can discuss a suitable alternative.

Don't hesitate to contact me directly with any questions (atosun@plos.org). 

Best regards, 

Alexandra 

Alexandra Tosun, PhD 

Associate Editor

PLOS Medicine

atosun@plos.org

Comments from editorial team:

In addition to our comments in the letter above, we ask you to consider the following comments:

1) The regimen is used in both first and second line therapy - what did the women who received other therapy receive, was this also endocrine/HER2 targeted? 

2) What is the standard of care and how does this regimen differ? 

3) Why was this particular endocrine therapy regimen chosen? How much influence did physician choice have on the treatment protocol?

4) The research question needs to be clarified and why it might be relevant in this treatment context. 

5) How does the choice of treatment regimen in your study compare with other settings worldwide?

Comments from the reviewers: 

Reviewer #1: The author evaluated the non-intravenous (also chemo-free) regimen for HR+/HER2+ breast cancer. The regimen has a very good response rate, long PFS, with clinical managable toxicity, which could provide a good clinical option in further clincial practice or research. This paper is well prepared. I would recommend "ACCEPT" for this paper after revision with 2 additional questions.

1. The explonary analysis of predictive factor for this regimen should be with caution due to its limited number of patients, especially for these patients with certain gene mutations (subgroup of a subgroup analysis). The mutaion rate of ERBB2 is very high for these patients receiving ctDNA sequecing analysis, is this HER2 amplification or mutation? 

2. Any pre-medication for this regimen, if has, the author needs to add this point in the manscript.

Reviewer #2: In this study, Zhang et al. aimed to evaluate the efficacy and safety of a chemotherapy-free regimen that combines dalpiciclib with pyrotinib and endocrine therapy for patients undergoing first- or second-line treatment for ER-positive and HER2-positive advanced breast cancer. Additionally, they conducted a biomarker analysis using 68Ga-HER2 PET/CT imaging and genomic testing. The study demonstrated promising results for a more convenient, purely outpatient treatment regimen. If validated in a phase III trial, it might become the new standard of care.

1.My main concern is the varying efficacy observed in similar yet distinct treatment regimens. In Yan Min's study, the median progression-free survival (PFS) for the dalpiciclib and pyrotinib combination is a modest 11 months, with the HR+ subgroup faring even worse at just 9.1 months. In Ouyang Quchang's study, however, the median PFS for the pyrotinib and endocrine therapy combination is more promising at 13.7 months. Moreover, in Wang Shusen's study focusing on the HER2+ brain metastasis patient population, the median PFS for the regimen comprising trastuzumab, pyrotinib, palbociclib, and endocrine therapy is a relatively lower 10.6 months. These treatment strategies, while different, exhibit a significant disparity in their efficacy.

2.The study screened 51 participants and enrolled 48 patients, of whom 3 were excluded: two because they did not meet the inclusion criteria, but the rest one the investigators did not specify why she were excluded.

3.In the Figure 1, is the "^" redundant?

Reviewer #3: Thanks for the opportunity to read your manuscript. My role is statistical reviewer, so I have focused on the design, data, and analysis that are presented. I have put general comments first, followed by questions relevant to a specific section of the manuscript (with a page/line reference). 

This manuscript presents a Phase II, single-arm study of patients with advanced HR+/HER2+ breast cancer, examining if the addition of CDK4/6 inhibitors to current treatment with anti-HER2/endocrine therapies shows evidence of efficacy. Patients were recruited from 6 large sites and includes patients with an ER2 and HER2 positive breast cancer. Treatment was based on data from a phase Ib study. Main study outcome was objective response rate (RECIST 1.1 guidelines), with a range of a secondary endpoints. The analysis was simple but appropriate for a single-arm design, with the estimates of the endpoints estimated with 95% confidence intervals (Clopper-Pearson for binary outcomes). ORR was 70.2%, with most patients achieving a partial response. Key components of the study match between the manuscript, protocol and study registration. 

I suspect it is unlikely to make a difference here, but simulation research suggests a liberal bias in the Clopper-Pearson binomial confidence intervals. For the main study outcome, could you confirm the interval is similar when calculated with a Wilson Score Interval calculation? 

Could a summary in the appendix be provided about the changes to the study protocol and the rationale for these from the first version to the final version? It looks like in the study registration there are less details about the age-based inclusion criteria (e.g. participants under <60 having additional inclusion criteria).

P3, L46. 'hypothesis generating trend' is a vague term, I think you could rephrase this to simply state these results instead.

P12, L227. I think it's ok to present the results by subgroup, but this should be carefully worded so that there is no direct comparison between them, as this study doesn't have the sample size to achieve this. 

Reviewer #4: The manuscript presents a Phase II study of an antiestrogen, an investigational CDK 4/6inhibitor and an investigational HER2 targeting TKI used as first or second line therapy in metastatic breast cancer that is hormone receptor positive and HER2 positive. 

The following comments are made in an attempt to improve the manuscript

The introduction and the discussion should address that in early line therapy of metastatic breast cancer the use of an AI and CDK 4/6 inhibitor has shown an overall survival improvement and the regimen of docetaxel, trastuzumab, and pertuzumab has shown an overall survival improvement. 

It would be of interest to know what standard of care practice is for patients with hormone receptor positive, HER2 positive metastatic breast cancer at the Institutions where patients were enrolled. 

The manuscript is overly enthusiastic over this 48 patient study used in a population that is expected to have somewhat similar outcomes with standard of care. The tone can be revised. 

The manuscript is inconsistent with use of ER Positive and Hormone Receptor Positive. It is suggested that a definition is provided and only one term is used throughout. 

The manuscript uses the term "resistance" in manners that does not apply to this patient population (first or second line therapy). Please modify the text. 

It is suggested that the analysis of the data be revised to only use those patients who met enrollment criteria. 

The discussion is too long and discusses matters not directly related to the data presented in this manuscript. The discussion can be shortened and made more focused.

---

* Please upload any figures associated with your paper as individual TIF or EPS files with 300dpi resolution at resubmission; please read our figure guidelines for more information on our requirements: http://journals.plos.org/plosmedicine/s/figures. While revising your submission, please upload your figure files to the PACE digital diagnostic tool, https://pacev2.apexcovantage.com/. PACE helps ensure that figures meet PLOS requirements. To use PACE, you must first register as a user. Then, login and navigate to the UPLOAD tab, where you will find detailed instructions on how to use the tool. If you encounter any issues or have any questions when using PACE, please email us at PLOSMedicine@plos.org.

* FINANCIAL DISCLOSURES: The funding statement should include: specific grant numbers, initials of authors who received each award, URLs to sponsors’ websites. Also, please state whether any sponsors or funders (other than the named authors) played any role in study design, data collection and analysis, the decision to publish, or preparation of the manuscript. If they had no role in the research, include this sentence: “The funders had no role in study design, data collection and analysis, decision to publish, or preparation of the manuscript.”

* COMPETING INTERESTS: All authors must declare their relevant competing interests per the PLOS policy, which can be seen here: https://journals.plos.org/plosmedicine/s/competing-interests

For authors with ties to industry, please indicate whether any of the interests has a financial stake in the results of the current study.

* The Data Availability Statement (DAS) requires revision. For each data source used in your study: 

* ETHICS STATEMENT: Please provide the approval number for each of the ethics committees.

FIGURES AND TABLES

SUPPLEMENTARY MATERIAL

REFERENCES

STUDY TYPE-SPECIFIC REQUESTS

* PLOS Medicine requires that all trials be prospectively registered in one of registries recognized by WHO. Please ensure that study registration details are included in the Methods section.

* Please structure the Methods section using the following sub-headings: Study design and participants, Randomization and masking, Procedures, Outcomes, Statistical analysis.

* Please ensure that all outcomes measures are reported according to the trial protocol [and/or trial registry]. Please clarify and explain all discrepancies between the paper and protocol. If the outcomes were not prespecified in the protocol, please define them in the Methods (Outcomes section) as post hoc and explain why they were added. Post-hoc comparisons should be presented as hypothesis generating rather than conclusive.

* Please ensure that all prespecified outcomes (primary, secondary, and exploratory) are listed in the Methods/Outcomes section and indicate whether there are outcomes that are not presented in the current report.

* Please specify the dates (Month Day, Year) during which study enrollment and follow up occurred.

* Please include absolute numbers wherever you report percentages; eg, n/N (%)

* Please present the safety data for the study including numbers of specific events and whether or not adverse events are thought to be related to treatment. AEs should be reported in the abstract, per CONSORT and CONSORT-Harms.

* Please complete the CONSORT checklist (https://www.equator-network.org/reporting-guidelines/consort/) and ensure that all components of CONSORT are present in the manuscript, including how randomization was performed, allocation concealment, blinding of intervention, definition of lost to follow-up, power statement. When completing the checklist, please use section and paragraph numbers, rather than page numbers.

* Please report your abstract according to CONSORT for abstracts, following the PLOS Medicine abstract structure (Background, Methods and Findings, Conclusions) https://www.equator-network.org/reporting-guidelines/consort-abstracts/

* If your trial had to undergo important modifications in response to extenuating circumstances, please complete the CONSERVE-CONSORT checklist and provide in your Supporting Information; (https://www.equator-network.org/reporting-guidelines/guidelines-for-reporting-trial-protocols-and-completed-trials-modified-due-to-the-covid-19-pandemic-and-other-extenuating-circumstances-the-conserve-2021-statement/). When completing the checklist, please use section and paragraph numbers, rather than page numbers.

* In keeping with our commitment to Open Science, please include the study protocol document and analysis plan (including any amendments) as Supporting Information to be published with the manuscript if accepted.

* Please note that PLOS Medicine requires prospective, public registration of a data sharing plan (as part of mandatory clinical trials registration) for all clinical trials that began enrollment on or after January 1, 2019, in accordance with ICMJE requirements.

* In the main text, please clarify how the data was analysed (intention-to-treat, per protocol, etc.) and explain all discrepancies between the manuscript and protocol.

---

## [Decision Letter · Decision Letter 2]

23 Jun 2025

Dear Dr. Zhang,

Thank you very much for re-submitting your manuscript "Dalpiciclib combined with pyrotinib and endocrine therapy in women with ER-positive, HER2-positive advanced breast cancer (PLEASURABLE): a multicentre, single-arm, phase 2 trial" (PMEDICINE-D-24-03621R2) for review by PLOS Medicine.

Thank you for your detailed response to the reviewers' and editors’ comments. I have discussed the paper with my colleagues and the academic editor, and it has also been seen again by three of the original reviewers. The changes made to the paper were satisfactory to the reviewers. Please carefully address the editors' comments below in a further revision. When submitting your revised paper, please once again include a detailed point-by-point response to the editorial comments.

The remaining issues that need to be addressed are listed at the end of this email. Any accompanying reviewer attachments can be seen via the link below. Please take these into account before resubmitting your manuscript: [LINK]

In revising the manuscript for further consideration here, please ensure you address the specific points made by each reviewer and the editors. In your rebuttal letter you should indicate your response to the reviewers' and editors' comments and the changes you have made in the manuscript. Please submit a clean version of the paper as the main article file. A version with changes marked must also be uploaded as a marked up manuscript file. Please also check the guidelines for revised papers at http://journals.plos.org/plosmedicine/s/revising-your-manuscript for any that apply to your paper.

We ask that you submit your revision within 1 week (Jun 30 2025). However, if this deadline is not feasible, please contact me by email, and we can discuss a suitable alternative.

Please do not hesitate to contact me directly with any questions (atosun@plos.org). If you reply directly to this message, please be sure to 'Reply All' so your message comes directly to my inbox.

We look forward to receiving the revised manuscript.

Sincerely,

Alexandra Tosun, PhD

Senior Editor 

PLOS Medicine

plosmedicine.org

Comments from Reviewers:

Reviewer #2: I have no other comments.

Reviewer #3: Thanks for the revised manuscript and responses to my original review. The explanation about the age-based inclusion criteria seems reasonable to me, noting that I am not a content expert. The changes to the manuscript and responses to my questions have resolved the queries from my initial review. 

Reviewer #4: The revision has improved the manuscript. The research is a Phase II study without impactful correlative science. The Discussion remains longer than needed.

[LINK]

Requests from Editors:

GENERAL

* Please confirm that your title complies with to PLOS Medicine's style. Your title must be nondeclarative and not a question. It should begin with main concept if possible. "Effect of" should be used only if causality can be inferred, i.e., for an RCT. Please place the study design ("A randomized controlled trial," "A retrospective study," "A modelling study," etc.) in the subtitle (i.e., after a colon).

* We suggest removing “PLEASURABLE” from the study title.

* Statistical reporting: Please revise throughout the manuscript, including tables and figures.

- Please report statistical information as follows to improve clarity for the reader ""22% (95% CI [13,28]; p</=)"".

- Please separate upper and lower bounds with commas instead of hyphens as the latter can be confused with reporting of negative values.

- Please repeat statistical definitions (HR, CI etc.) for each set of parentheses.

* Please ensure that all abbreviations are defined at first use throughout the text (including statistical abbreviations). Please also check figures and tables.

* Please ensure that tables and figures, including those in supplementary files, are appropriately referenced in the main text.

* Please check that any use of statistical terms (such as trend or significant) are supported by the data, and if not please remove them.

* Data availability: The current statement requires revision. Please note that the de-identified datasets should not have a time limit for access. PLOS Medicine requires that the de-identified data underlying the specific results in a published article be made available, without restrictions on access, in a public repository or as Supporting Information at the time of article publication, provided it is legal and ethical to do so. If the data are not freely available, please describe briefly the ethical, legal, or contractual restriction that prevents you from sharing it. Please also include an appropriate contact (web or email address) for inquiries; this cannot be a study author.

* We strongly encourage you to have a native English speaker edit the manuscript. We still think the manuscript needs to clarify what the specific novelty is. For example, is it the use as frontline therapy or testing a different dose?

* Please ensure that you have correctly reported the pre-specified outcomes and post-hoc analyses according to the trial protocol.

* Please complete the CONSORT checklist with the relevant details for your study. We appreciate that not maybe all all components might be relevant. Please upload the completed checklist as a supporting information file and include a reference in the methods section.When completing the checklist, please use section and paragraph numbers, rather than page numbers.

ABSTRACT

* Please confirm that your abstract complies with our requirements, including providing all the information relevant to this study type https://journals.plos.org/plosmedicine/s/submission-guidelines#loc-abstract

* Per CONSORT, please note that only the primary outcome of the trial should be reported in your Abstract. Secondary outcomes should only be included in the Abstract if all secondary outcomes are fully reported. For trials that have many secondary outcomes, the Abstract should be limited to reporting the primary outcome.

* Why aren't you reporting plasma pharmacokinetics as a secondary outcome?

* Please include population and setting, years during which the study took place and length of follow up.

* Please state that analysis was intention to treat.

* Please provide the number of participants lost to follow up in each group.

* Please ensure that all numbers presented in the abstract are present and identical to numbers presented in the main manuscript text.

* Please report the variance (ie 95%CI values) for DCR.

* Please remove the word ‘promising’.

* Please clarify why you describe the primary outcome “ORR” as “investigator confirmed ORR” in the results section of the Abstract.

AUTHOR SUMMARY

* In the author summary, in the final bullet point of 'What Do These Findings Mean?', please include the main limitations of the study in non-technical language.

INTRODUCTION

* Please ensure that the Introduction ends with a clear description of the study question or hypothesis.

* Please remove the word ‘impressive’.

METHODS AND RESULTS 

* It appears that the disease control rate (DCR) is reported as a secondary outcome in the main text, yet it is not mentioned in the trial protocol. If DCR was not a pre-specified secondary outcome, please report it as a post hoc analysis.

* Please clarify the time points of collection for the different analyses conducted as part of the biomarker analysis.

* Please confirm that the HER2-PET assessment described on page 13 of the trial protocol describes the 68Ga-HER2 affibody with PET-CT descrbied in the main text and explain why the method was not further defined in the trial protocol.

* In the Methods section, please report the amendments undertaken between the original trial protocol (Version 1.0) and the final trial protocol (Version 3.0). For transparenty, we encourage you to provide a file outlining the amendments.

* ”31 (65.6%) had received previous HER2-target therapy and 37 (77.1%) patients had received previous endocrine therapy in any setting.” – according to Table 1, 31 patients equals 64.6% that had received previous HER2-target therapy. Please check that the numbers in the main text are correct.

* “Overall survival (OS) data was immature with a total of 7 events, and the estimated 48-month OS rates were 82.6% (95% CI, 66.3 to 91.5) by the data cutoff date (S1 Fig).” – According to the trial protocol, overall survival was not a pre-specified outcome. Please report as a post-hoc analysis.

* S2Fig: We believe you should mention the very wide confidence intervals for the subgroup results which indicate a high degree of uncertainty.

* ”The median PFS in patients with first-line and second-line HER2-targeted therapy was 22.3 months (95% CI, 16.6 to 26.6) and 20.1 months (95% CI, 7.6 to 44.1), respectively (Fig 3).” – Please note that you report the median PFS already on line 237. Please improve reporting and avoid repetition.

* How many samples were included in the pharmacokinetics analysis?

* According to the protocol, blood plasma samples were planned to be collected from 8-10 participants on day 21. In the main text, you state that 16 patients had baseline ctDNA submitted for testing. Please explain the discrepancy.

* How were the patient numbers determined for the 68Ga-HER2 and 18F-FDG PET/CT scans?

* Was Next Generation Sequencing a pre-specified analysis? We cannot find any information on this in the protocol, only on the collection of tumor tissue samples. Please explain and report as a post-hoc analysis if accurate.

DISCUSSION

* Pleas remove all subheadings.

* When revising the discussion, please consider toning down any conclusions given the small study size and the lack of a comparator group.

* Please remove the word ‘promising’.

General Editorial Requests

---

## [Editor Report · Decision Letter 3]

2 Jul 2025

Dear Dr Zhang, 

On behalf of my colleagues and the Academic Editor, Ricky W Johnstone, I am pleased to inform you that we have agreed to publish your manuscript "Dalpiciclib combined with pyrotinib and endocrine therapy in women with ER-positive, HER2-positive advanced breast cancer: a prospective, multicentre, single-arm, phase 2 trial" (PMEDICINE-D-24-03621R3) in PLOS Medicine.

I appreciate your thorough responses to the reviewers' and editors' comments throughout the editorial process. We look forward to publishing your manuscript, and editorially there are only a few remaining points that should be addressed prior to publication. We will carefully check whether the changes have been made. If you have any questions or concerns regarding these final requests, please feel free to contact me at atosun@plos.org.

Please see below the minor points that we request you respond to:

1) Data availability: Please update the online submission form with the details provided on lines 1443-1445 (track changes version). We suggest the following statement: “All relevant de-identified data supporting the findings of this study are included within the manuscript and its Supporting Information files. The raw clinical data are protected due to privacy laws and unavailable.”

2) Trial protocol: According to your rebuttal, the protocol you have provided is a translated version. Please also provide the trial protocol in its original language. Both the original language and translated versions of the trial protocol will be published alongside the manuscript to ensure transparency.

3) Methods/Results: “A total of 12 participants provided samples for the pharmacokinetics (PK) analysis.” – please include this information in the main text.

Before your manuscript can be formally accepted you will need to complete some formatting changes, which you will receive in a follow up email (including the editorial points above). Please be aware that it may take several days for you to receive this email; during this time no action is required by you. Once you have received these formatting requests, please note that your manuscript will not be scheduled for publication until you have made the required changes.

PRESS

Sincerely, 

Alexandra Tosun, PhD 

Senior Editor 

PLOS Medicine